# MiRNA-21a, miRNA-145, and miRNA-221 Expression and Their Correlations with WNT Proteins in Patients with Obstructive and Non-Obstructive Coronary Artery Disease

**DOI:** 10.3390/ijms242417613

**Published:** 2023-12-18

**Authors:** Alfiya Oskarovna Iusupova, Nikolay Nikolaevich Pakhtusov, Olga Alexandrovna Slepova, Yuri Nikitich Belenkov, Elena Vitalievna Privalova, Irina Vladimirovna Bure, Ekaterina Alexandrovna Vetchinkina, Marina Vyacheslavovna Nemtsova

**Affiliations:** 1Department of Hospital Therapy No 1, I.M. Sechenov First Moscow State Medical University (Sechenov University), 119991 Moscow, Russiaslepova_o_a@staff.sechenov.ru (O.A.S.); belenkov_yu_n@staff.sechenov.ru (Y.N.B.);; 2Laboratory of Medical Genetics, I.M. Sechenov First Moscow State Medical University (Sechenov University), 119991 Moscow, Russia; bureira@mail.ru (I.V.B.); katevetchinkina@ya.ru (E.A.V.); nemtsova_m_v@mail.ru (M.V.N.); 3Research Institute of Molecular and Personalized Medicine, Russian Medical Academy of Continuous Professional Education, 125445 Moscow, Russia; 4Laboratory of Epigenetics, Research Centre for Medical Genetics, 115522 Moscow, Russia

**Keywords:** microRNA (miR-21a, miR-145, miR-221), WNT cascade proteins (WNT1, WNT3a, WNT4, WNT5a, LRP6), SIRT1, ischemia with no obstructive coronary arteries, coronary artery disease, cardiovascular diseases

## Abstract

MicroRNAs and the WNT signaling cascade regulate the pathogenetic mechanisms of atherosclerotic coronary artery disease (CAD) development. Objective: To evaluate the expression of microRNAs (miR-21a, miR-145, and miR-221) and the role of the WNT signaling cascade (WNT1, WNT3a, WNT4, and WNT5a) in obstructive CAD and ischemia with no obstructive coronary arteries (INOCA). Method: The cross-sectional observational study comprised 94 subjects. The expression of miR-21a, miR-145, miR-221 (RT-PCR) and the protein levels of WNT1, WNT3a, WNT4, WNT5a, LRP6, and SIRT1 (ELISA) were estimated in the plasma of 20 patients with INOCA (66.5 [62.8; 71.2] years; 25% men), 44 patients with obstructive CAD (64.0 [56.5; 71,0] years; 63.6% men), and 30 healthy volunteers without risk factors for cardiovascular diseases (CVD). Results: Higher levels of WNT1 (0.189 [0.184; 0.193] ng/mL vs. 0.15 [0.15–0.16] ng/mL, *p* < 0.001) and WNT3a (0.227 [0.181; 0.252] vs. 0.115 [0.07; 0.16] *p* < 0.001) were found in plasma samples from patients with obstructive CAD, whereas the INOCA group was characterized by higher concentrations of WNT4 (0.345 [0.278; 0.492] ng/mL vs. 0.203 [0.112; 0.378] ng/mL, *p* = 0.025) and WNT5a (0.17 [0.16; 0.17] ng/mL vs. 0.01 [0.007; 0.018] ng/mL, *p* < 0.001). MiR-221 expression level was higher in all CAD groups compared to the control group (*p* < 0.001), whereas miR-21a was more highly expressed in the control group than in the obstructive (*p* = 0.012) and INOCA (*p* = 0.003) groups. Correlation analysis revealed associations of miR-21a expression with WNT1 (r = −0.32; *p* = 0.028) and SIRT1 (r = 0.399; *p* = 0.005) protein levels in all CAD groups. A positive correlation between miR-145 expression and the WNT4 protein level was observed in patients with obstructive CAD (r = 0.436; *p* = 0.016). Based on multivariate regression analysis, a mathematical model was constructed that predicts the type of coronary lesion. WNT3a and LRP6 were the independent predictors of INOCA (*p* < 0.001 and *p* = 0.002, respectively). Conclusions: Activation of the canonical cascade of WNT-β-catenin prevailed in patients with obstructive CAD, whereas in the INOCA and control groups, the activity of the non-canonical pathway was higher. It can be assumed that miR-21a has a negative effect on the formation of atherosclerotic CAD. Alternatively, miR-145 could be involved in the development of coronary artery obstruction, presumably through the regulation of the WNT4 protein. A mathematical model with WNT3a and LRP6 as predictors allows for the prediction of the type of coronary artery lesion.

## 1. Introduction

According to the World Health Organization, coronary artery disease (CAD) occupies a leading position among the ten main causes of death in the world (2019) [1]. The morbidity of CAD continues to remain at an extremely high level. In 2020, about 244.1 million people worldwide suffered from CAD, and the death rate was 8.95 million [2]. The high prevalence of CAD in the population is caused by many well-known factors. The epigenetic component is of particular interest among them because of its important role in the regulation of atherosclerotic lesions of the coronary arteries (CA) [3,4]. The effect of the WNT/β-catenin cascade on the progression of atherosclerosis has been confirmed in several studies, and the regulation of arterial calcification mediated by the WNT signaling pathway additionally confirms its participation in the coronary lesions and the development of CAD [5,6].

The WNT cascade consists of 19 genes encoding lipoproteins of the WNT family [7] and is divided into canonical and non-canonical pathways. The non-canonical WNT pathway is independent of the β-catenin factor. In contrast, the canonical WNT pathway involves the translocation of β-catenin into the nucleus with subsequent activation of various target genes of such protein transcription factors as TCF, LEF (T-cell factor/lymphoid enhancer factor), and others [8,9]. The canonical WNT pathways (for example, WNT1 and WNT3a) mainly control cell proliferation and differentiation of myofibroblasts, thereby contributing to fibrogenesis. In contrast, the non-canonical WNT pathways (for example, WNT4 and WNT5a) regulate cell polarity and migration [10,11]. In vivo studies have established the role of the WNT pathway at all stages of atherosclerosis development. However, much of its pathogenesis remains poorly described or contradictory [12]. Wnt1 induces cardiac fibroblast proliferation and expression of pro-fibrotic genes. WNT4 promotes smooth muscle cell proliferation and migration. WNT1, WNT4, and WN7A expression significantly increased after myocardium infarction [13]. It was experimentally demonstrated that WNT5a limits the deposition of cholesterol in the arterial walls by activating the mechanism of its reverse transport [14]. Recent studies have demonstrated the important role of WNT5A in the regulation of endothelial permeability [15]. WNT5a suppresses vascular smooth muscle cell (VSMC) apoptosis caused by oxidative stress by inducing WNT1-inducible secreted protein-1 (WISP-1) [16]. WNT3A has an anti-inflammatory effect by suppressing GSK3ß. Both WNT3A and WNT5A have been shown to activate nuclear transcription factor (NF-KB) in mouse and human macrophages [17]. WNT5A activates β-catenin-independent signaling in endothelial cells (EC) and enhances angiogenesis by increasing their proliferation and survival [18,19].

Low-density lipoprotein receptor-related protein 6 (LRP6) is a member of the low-density lipoprotein receptor (LDLR) family and influences the process of lipoprotein endocytosis [20]. Phosphorylation of LRP6 is involved in the activation of WNT/β-catenin signaling [21]. Patients with the LRP6-R611C mutation had extremely high plasma cholesterol levels, and common variants of the LRP6 gene were also associated with a moderate increase in LDL cholesterol levels in plasma in the general population. LRP6 provides key protection against dyslipidemia and atherosclerosis [22]. According to several studies, there is a negative correlation between LRP6 and the expression of various miRNAs. Thus, ectopic overexpression of miR-21 by antagomirs (synthetic RNAs that are completely complementary to specific miRNA targets and, therefore, can inactivate them) leads to increased LRP6 protein levels. That causes the activation of the WNT signaling pathway and consequently decreases inflammation and lipid deposition in the liver [23].

The expression of most components involved in signal transmission through the WNT cascade is controlled by microRNAs (miRNAs), resulting in highly dynamic post-transcriptional regulation, and its aberrations ultimately lead to the development of diseases [24]. MiRNAs can act as diagnostic markers of CVD associated with atherosclerosis [25]. According to the research data, several of them have already been identified. For example, downregulation of miR-21 in macrophages enhances the expression of proinflammatory cytokines TNF-a (tumor necrosis factor-alpha), IL-6 (interleukin-6), and IL-1b (interleukin-1 beta). Moreover, protein levels of COX-2 (cyclooxygenase-2) and the induced isoform of nitric oxide synthase (iNOS) were increased in macrophages of bone marrow origin (BMDM) with miR-21 deficiency after their treatment with lipopolysaccharides. Thus, miR-21 deficiency in hematopoietic cells contributes to the progression of atherosclerosis and enhances plaque remodeling with thinning of the fibrous capsule, which clinically leads to an increased number of adverse cardiovascular events [26]. The other study revealed that miR-21 enhances the adhesion of monocytes to endothelial cells and, consequently, leads to the development of inflammation of the vascular wall [27]. It confirms the involvement of miR-21 in proinflammatory and proatherosclerotic processes through activation of VCAM-1 (vascular cellular adhesion molecule-1), ICAM-1 (intercellular adhesion molecule 1), and MCP1 (monocyte chemotactic protein-1) by affecting various signaling pathways [28]. Therefore, the role of miR-21 in the pathogenesis of atherosclerosis remains controversial.

MiRNAs could be one of the most promising epigenetic markers not only for diagnosis but also for determining the prognosis of CAD. MiRNAs are a class of small non-coding RNAs of 18–25 nucleotides in length [29,30]. They can regulate the development of CAD by regulating various processes, such as modulation of angiogenesis (miR-92a-3p, miR-939, and miR-206), inflammatory reactions (miR-181a-5p, miR-181a-3p, miR-216a, and miR-383-3p), leukocyte adhesion (miR-21 and miR-25), and vascular smooth muscle cell (VSMC) activity (miR-574-5p) [11].

Sirtuin-1 (SIRT1) is a class III histone/protein deacetylase [31]. Its overexpression can increase cell proliferation and significantly suppress apoptosis. Overexpression of SIRT1 has been confirmed to inhibit the contraction and proliferation of VSMC [32]. According to the conducted studies, SIRT1 increases nitric oxide (NO) production by activating endothelial nitric oxide synthase (eNOS), which leads to vasodilation and a decrease in endothelial dysfunction degree [33,34]. Overexpression of miR-34a and miR-217 contributes to the progression of endothelial dysfunction by suppressing SIRT1 [35,36]. SIRT1 is involved in the regulation of oxidative stress processes. Also, it suppresses inflammation in the vascular wall by reducing the activity of NF-κB (nuclear factor kappa-light-chain-enhancer of activated B cells) [37]. In the experimental model, it was established that SIRT1 regulates autophagy and apoptosis of ischemic cardiomyocytes by activating AMP-activated protein kinase (AMPK) and also suppresses the production of foam cells and, thereby, prevents the progression of atherosclerosis [38].

It seems promising to conduct a study on the epigenetic regulation of the WNT signaling cascade by miRNAs in the cardiovascular system, including atherosclerosis of CA, which eventually leads to the development of severe complications (acute myocardial infarction, left ventricular aneurysm, acute and chronic heart failure, sudden cardiac death). In this study, we address the role of miR-21a, miR-145, and miR-221 and their correlations with protein levels of some members of the WNT and SIRT1 cascade.

According to the literature, there is limited data on miRNA expression in patients with CAD, including ischemia with no obstructive coronary arteries (INOCA) [39,40]. It should be noted that in these studies, the concentrations of proteins of the WNT signaling pathway in patients with non-obstructive CAD were not additionally studied, and the analysis of the possible relationship between the expression of miRNAs and WNT proteins, SIRT1 and LRP6 in these groups of patients was not carried out.

The aim of the study: Epigenetic regulation of WNT-family proteins by miRNA-21a, miRNA-145, and miRNA-221 can mediate coronary artery disease pathogenesis. We aimed to evaluate the levels of these miRNAs, WNT-proteins (WNT1, -3a, -4, -5a), and LRP6 in adults with coronary artery disease against a control population.

## 2. Results

### 2.1. Basic Clinical Characteristics

All plasma samples were obtained from patients hospitalized in the cardiology department with chest pain and shortness of breath who have undergone physical activity tests (stress Echo-CG, myocardial scintigraphy, and cardiac MRI with stress test) to verify the diagnosis of CAD. Patients with confirmed myocardial ischemia further underwent imaging of the CA (coronary angiography or MSCT of the coronary arteries) to address the need for their revascularization. The study included 64 patients, who were divided into two groups depending on the degree of CA obstruction: group 1—patients with hemodynamically insignificant stenosis (INOCA, stenosis < 50%), and group 2—patients with CA obstruction (obstructive CAD, stenosis > 50%). The third group (control) comprised 30 healthy volunteers without CVD risk factors. The design of the study is presented in Figure 1.

The general clinical and demographic characteristics of the groups are summarized in Table 1. The investigated groups differed in age and BMI from the control group. In the group of patients with INOCA, women prevailed in a ratio 3:1 (75% women vs. 25% men).

All patients received the recommended therapy according to the national and international clinical guidelines (Table 2). The differences in total cholesterol and low-density lipoproteins (LDL) levels between the groups are most likely determined by the better results in reducing the level of total cholesterol and LDL in patients with obstructive CAD who were treated with higher doses of statins. The analysis of the daily dose of statins revealed statistically significant differences (*p* = 0.02) between the groups depending on the type of CA lesions.

### 2.2. Concentration of WNT Proteins in Plasma

According to the results of the study, significant differences in the concentration of WNT1, WNT3A, WNT4, WNT5a, and SIRT 1 were revealed not only when compared with the control group but also between the variants of CADs (obstructive and non-obstructive). When analyzing the results in the groups, higher levels of WNT1 and WNT3a proteins were noted in patients with CAD and obstructive CA lesions. The obstructive CAD group was markedly lower in WNT4 and WNT5a expression versus the INOCA and control groups, which were almost identical. The level of SIRT1 was significantly higher in the INOCA group (as well as in the control group) versus oCAD. The concentration of LRP6 in the study groups did not differ significantly (Table 3).

### 2.3. MiRNA Expression in Plasma of Patients with CAD

The expression level of miR-221 was significantly higher in all patients with CAD (obstructive and non-obstructive) in comparison with the control group (*p* < 0.001). In contrast, the expression of miR-21a was significantly higher in the control group than in the obstructive CAD (*p* = 0.012) and INOCA (*p* = 0.003) (Figure 2). The expression level of miR-145 was not significantly different between the groups (*p* = 0.069).

### 2.4. Correlations of WNT Proteins with Circulating miRNAs

Correlation analysis revealed weak associations between miR-21a expression and protein levels of WNT1 (r = -0.32; *p* = 0.028) and moderate linkage with SIRT1 (r = 0.399; *p* = 0.005) in patients with stable CAD (all CAD groups). MiR-145 expression was correlated with WNT4 protein concentration in patients with CAD and hemodynamically significant CA stenosis (r = 0.436; *p* = 0.016). However, no significant correlations between miRNA expression and WNT proteins were determined in the group of INOCA.

Univariate logistic regression determined concentrations of LRP6 and WNT3a as significant predictors of INOCA. The results of univariate logistic regression are presented in Table 4.

Multivariate regression analysis allowed for obtaining a mathematical model predicting the type of coronary lesion. Table 5 demonstrates that WNT3a and LRP6 are significant independent predictors of the degree of CA obstruction.

This ROC curve predicts coronary lesions in patients with stable CAD (cut-off = 0.35) with sensitivity 83.3% [66.7%; 95.5%], specificity 83.3% [71.0%; 93.3%], ROC-AUC = 92.6% [82.9%; 99.0%]. If the cut-off is >0.35, then we can assume the probability of obstructive CAD (Figure 3).

## 3. Discussion

Atherosclerosis of the CA is a multifactorial disease, and epigenetic factors regulating signal transmission through the WNT cascade play a significant role in its development and progression. The involvement of the WNT signaling cascade has been established in the pathogenesis of atherosclerosis in all stages. Dysregulations of the WNT signaling during oxidative stress and/or inflammation may be a common molecular mechanism contributing to the development of atherosclerosis, insulin resistance, and hyperlipidemia, whose frequencies increase with age [8].

In the present study, the highest concentrations of WNT1 and WNT3a proteins were determined in a group of patients with obstructive CAD. According to the results of Wang et al., inhibition of WNT1 signaling by SIRT6 promotes lipophagy and increases plaque stability [41]. At the same time, according to Brown et al., the WNT3a protein is expressed in atherosclerotic plaques of human CA. It is actively involved in the inhibition of VSMC apoptosis induced by oxidative stress [42]. Thus, higher levels of WNT1 and WNT3a may be associated with the development of significant stenosis of CA. In addition, according to the studies, the upregulation of WNT4 under the influence of platelet-derived growth factor-BB (PDGF-BB) promotes VSMC proliferation through the frizzled-1 receptor and β-catenin, which explains the close correlation between WNT4 and the progression of stenosing atherosclerosis [43,44]. According to the results of our study, a correlation between WNT5a protein and the levels of total cholesterol and LDL was revealed.

MiRNAs are involved in the regulation of various members of the WNT cascade, and, on the other hand, the expression of miRNAs themselves is regulated with the participation of the WNT signaling cascade [45].

According to our data, miR-21a expression was correlated with the level of SIRT1 in patients with CAD (obstructive and INOCA). However, these variables were inversely proportional in the control group. The SIRT1 participates in a cascade of reactions that prevent excessive endothelium damage under the influence of oxidative stress and inflammation and thus has cardioprotective properties. Finally, SIRT1 reduces oxidative stress, which is one of the most important factors contributing to the development of atherosclerosis [46]. According to experimental data, overexpression of SIRT1 from low (2.5-fold) to moderate (7.5-fold) prevents myocardial hypertrophy, development of apoptosis/fibrosis, and heart failure and decreases expression of aging markers. However, the high level of SIRT1 (12.5-fold) promotes apoptosis, hypertrophy, and decreased contractile function of the myocardium. Thus, it is possible that moderate expression of SIRT1 induces myocardial resistance to oxidative stress and apoptosis and has a protective effect, whereas high concentrations have the opposite effect in vivo [36]. Based on the data of our investigation and the available results of other studies obtained mainly experimentally, it would be difficult to unambiguously define the role of miR-21a expression and the level of SIRT1 in CAD. Therefore, further investigations and the commencement of large-scale fundamental research are needed.

A number of miRNAs regulate the differentiation of cardiomyocytes in cardiac and mesenchymal stem cells by modulating the expression of sFRP2 (secreted frizzled-related protein 2) and β-catenin, respectively [46,47,48]. Being involved in the process of atherogenesis, miRNAs can also act as diagnostic markers of CVD associated with atherosclerosis [25]. Bazan et al. have confirmed the role of circulating miR-221 and miR-222 in the thickening of intima, which is the initial stage of atherosclerotic plaque formation. Thus, decreased expression of miR-221 and miR-222 may lead to inhibition of VSMC proliferation and, consequently, to thinning of the fibrous covering of the atherosclerotic plaque and its damage [49]. In our work, we determined significantly higher miR-221 expression in patients with obstructive CAD and INOCA than in the control group. That is understandable, given the role of this miRNA in the processes of atherogenesis.

The current study revealed a positive correlation between miR-145 expression and WNT4 protein level in a group of patients with obstructive CA lesions. Similar results were obtained by Knoka et al. in 2020. Thus, they confirmed an association of miR-145 expression with the increase of the necrolipid nucleus of an atherosclerotic plaque in patients with stable CAD [50]. In addition, it is known that there is a close correlation between the WNT4 protein level and the development of stenosing atherosclerosis. It should be noted that alternative results were also obtained. Thus, according to the investigation of Gao et al., lower expression of miR-145 causes multivessel CA lesions in patients with CAD [51].

Based on the results of the ROC analysis, O’Sullivan et al. concluded that four miRNAs (miR-15a-5p, miR-146a-5p, miR-16-5p, and miR-93-5p) were predictors of the stable CAD development [52]. Zhang et al. suggested other miRNAs (miR-29a-3p, miR-574-3p, and miR-574-5p with AUCs 0.830, 0.792, and 0.789, respectively) as potential markers for noninvasive diagnosis of CAD [53]. Our study found no potential non-invasive diagnostic markers of stable CAD among investigated miRNAs. It may be related to the insufficient sample size. Therefore, the study on a larger number of samples is still required.

Today, there is an increasing amount of data on the efficacy of targeting miRNAs in the treatment of diseases associated with atherosclerosis. MiRNAs could also be potential targets for CAD therapy [54,55]. Some members of the WNT signaling cascade are also attractive targets for therapeutic intervention by either low-molecular inhibitors or biological drugs that mimic or modulate the components of extracellular regulation.

Based on the results of the multivariate regression analysis, we can assume WNT3a and LRP6 are independent predictors of the type of CA lesion in patients with stable CAD. Therefore, these components of the WNT and LRP6 signaling cascade may be potential diagnostic biomarkers of stenosing atherosclerosis.

Our first results can be useful in comprehending the potential roles of miRNA. We plan to continue further studies on the mechanisms of epigenetic regulation of the WNT signaling pathway by miRNA, which may be associated with obstructive atherosclerosis of CA. The development of novel strategies for the successful drug treatment of CAD patients with different variants of CA lesions seems promising. Thus, this issue requires further careful study and fundamental research.

## 4. Materials and Methods

### 4.1. Patient Population

A cross-sectional observational study included 94 subjects, who were eligible according to the inclusion criteria and signed informed consent from 2020 to 2022. The study was conducted in accordance with the Declaration of Helsinki and approved by the Ethics Committee of the Sechenov University (Number: 01–21, 22 January 2021). The study included men and women aged 45–75 years with a verified CAD diagnosis.

The myocardial ischemia in hospitalized patients with stable angina or its analogs was confirmed by using instrumental diagnostic methods, namely stress echocardiography (echo-CG) or single-photon emission computed tomography (myocardial scintigraphy), against the background of the exercise testing. Depending on the results of coronary angiogram (CAG) or multispiral computed tomography (CT), the patients were divided into two groups: 20 patients with non-obstructive CA lesions (stenosis <50% or unchanged CA); 44 patients with obstructive CAD (presence of hemodynamically significant CA stenosis). The control group (*n* = 30) included healthy volunteers without CVD and risk factors. Exclusion criteria were the following: diabetes mellitus, acute coronary syndrome, myocardial infarction, and stroke in the previous 3 months, chronic heart failure III-IV functional class (NYHA), autoimmune and oncological diseases, signs and symptoms of liver disease in the decompensation stage, portal hypertension, uncontrolled bronchial asthma and chronic obstructive pulmonary disease, gastric or duodenal ulcer in the exacerbation stage, chronic pancreatitis in the exacerbation stage, malignant neoplasms, thyroid diseases, Cushing’s syndrome, acute renal failure, terminal renal failure (GFR < 15 mm/min/1.73 m^2^), mental illness, alcoholism, drug addiction, substance abuse, pregnancy, and breastfeeding.

### 4.2. Collection of Blood Samples and ELISA

Blood plasma samples were collected in tubes with EDTA K3 as an anticoagulant, centrifuged for 20 min at 1000× *g* (1000 RCF), and further frozen in cryotubes at −80 °C. To estimate the WNT protein levels, LRP6 and SIRT1 enzyme immunoassay (ELISA) was performed on the ELISA analyzer Adaltis Personal Lab (Rome, Italy) using Cloud-Clone Corp. kits (Wuhan, CCC, USA) (catalogue numbers: SEL821Hu, SEL817Hu, SEP155Hu, SED105Hu, SEP549Hu, SEE912Hu). The coefficient of variation (CV) for the sets was 10% and 12%, respectively.

All patients have undergone standard biochemical tests, including indicators of the lipid spectrum, glucose, and uric acid.

### 4.3. RNA Extraction and Reverse Transcription-Polymerase Chain Reaction (RT-PCR) Assay

Blood total RNA, including miRNA, was extracted from samples using Qiazol (Qiagen, Germany) following the manufacturers’ protocols. The concentration and purity of the obtained RNA were estimated on the NanoDrop 2000 microvolume spectrophotometer (Thermo Fisher Scientific, Waltham, MA, USA). The process of extraction was repeated for each sample until a sufficient amount of RNA was obtained for the next steps.

cDNA was synthesized using MiScript II RT Kit (Qiagen, Hilden, Germany) according to the recommended protocol. To obtain cDNA, 300 ng of total RNA isolated from each sample was used.

The expression level for each analyzed miRNA and the control was quantified in three repetitions on the CFX96 Real-Time PCR Detection System (Bio-Rad, Hercules, CA, USA) by using the MiScript SYBR Green PCR Kit (Qiagen, Hilden, Germany; Catalogue number: 218073) according to the manufacturer’s recommended program (15 min at 95 °C, followed by 40 cycles of 94 °C for 15 s, 55 °C for 30 s, and 70 °C for 30 s). Primers for miRNAs were designed according to the instructions [56], and their sequences are listed in Table 6. The presynthesized MiScript Primer Assay (Ce_miR-39_1, identification code MS00019789, Qiagen, Germany) was used for the control. The obtained Ct values were normalized to the exogenous control cel-miR-39-3p and analyzed using the 2^−ΔCt^ method. The results are presented as REU (relative units of expression).

### 4.4. Statistical Analysis

Statistical analysis of the results was performed using the program StatTech v.v. 3.1.10 (StatTech, Russia) and the free Python computing software environment (v.3.11). The normality of sample distribution was evaluated using the Shapiro–Wilk (*n* < 50) or Kolmogorov–Smirnov (*n* > 50) tests. Quantitative variables with normal distribution were described using arithmetic averages (M) and standard deviations (SD) with a 95% confidence interval (95% CI). When the distribution of variables differed from the normal, quantitative data were described using the median (Me) and the lower and upper quartiles (Q1; Q3). The two groups were compared quantitatively with an abnormal distribution using the Mann–Whitney U-test. Three or more groups were compared quantitatively with an abnormal distribution using the Kruskal–Wallis test; the post-hoc testing was performed using the Dunn test with Bonferroni adjustment. To estimate the diagnostic significance of quantitative variables in predicting a certain outcome, the ROC curve analysis method was used. The cut-off value of the quantitative variable was determined to correspond to the maximum Youden index.

Multiple logistic regression (MLR) was used to build a model for predicting the presence/absence of a characteristic. The choice of the method was based on the dichotomy of the dependent variable and the fact that independent variables characterize both categorical and quantitative characteristics. The independent variables were selected by step-by-step reverse selection using Wald statistics as an exclusion criterion. The statistical significance of the obtained model was determined using the criterion χ2. To estimate the quality of the constructed model, the following criteria were used: ROC-AUC, accuracy, sensitivity, specificity, and DCA analysis (decision curve analysis). Metrics were calculated together with 95% CI. The 95% CI was calculated using the bootstrap method with a sample of 1000 instances. The threshold value was chosen in accordance with the maximization of sensitivity and specificity.

## 5. Conclusions

In patients with obstructive CAD, higher levels of WNT1 and WNT3a proteins, which are part of the canonical WNT-β-catenin pathway, were found. In contrast, the concentrations of WNT4 and WNT5a proteins belonging to the non-canonical WNT-β-catenin pathway were higher in the INOCA and control groups. According to our data, the expression level of miR-21a positively correlated with the level of SIRT1 and negatively correlated with WNT1 activity. Therefore, we can assume this microRNA’s possible contribution to the atherosclerosis process in CAD. MiR-145 was positively correlated with WNT4, with a decrease in the expression of which the progression of atherosclerosis has been proven according to literature data.

The multivariate regression analysis allowed us to obtain a model that can predict the type of CA lesion with high sensitivity and specificity. WNT3a and LRP6 can be used as predictors.

## Figures and Tables

**Figure 1 ijms-24-17613-f001:**
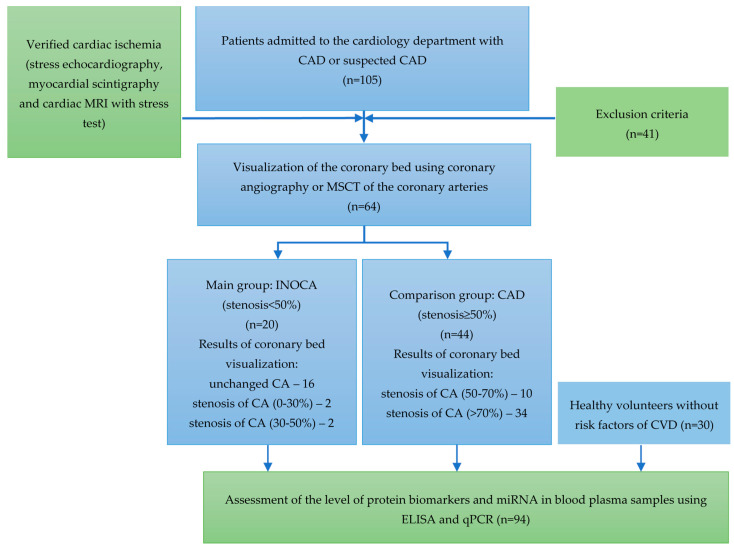
The design of the study.

**Figure 2 ijms-24-17613-f002:**
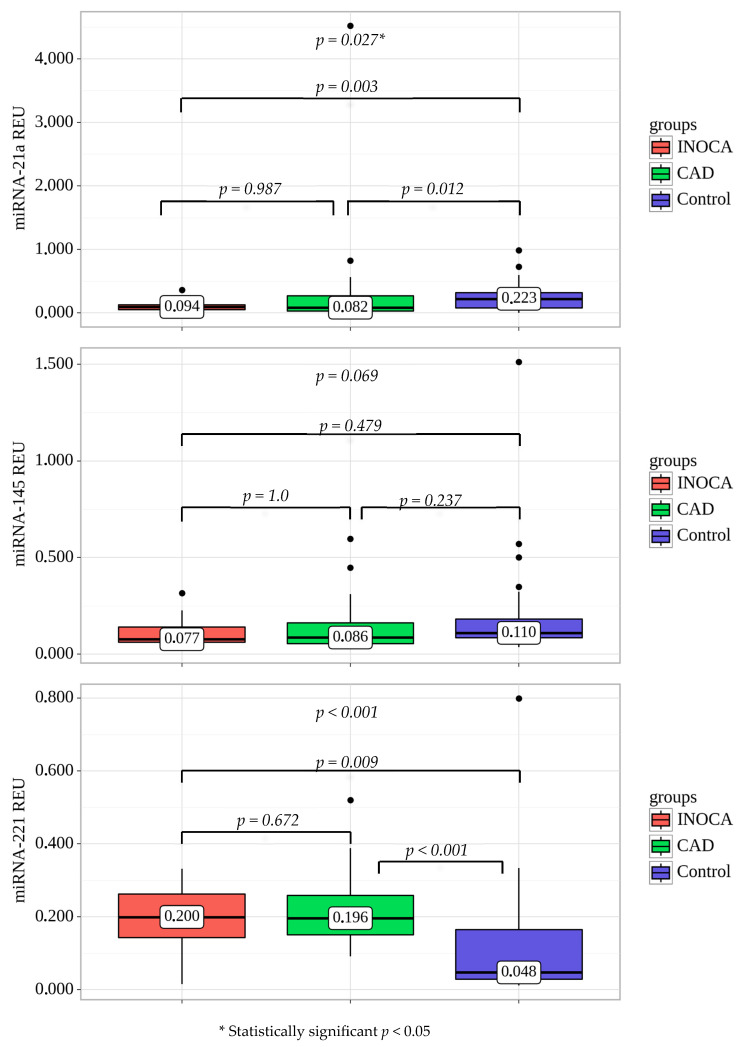
MiR expression in plasma of CAD patients and healthy volunteers (control). All values are presented as the median and CI. Statistically significant *p* < 0.05; CAD—coronary artery disease, INOCA—ischemia with no obstructive coronary arteries.

**Figure 3 ijms-24-17613-f003:**
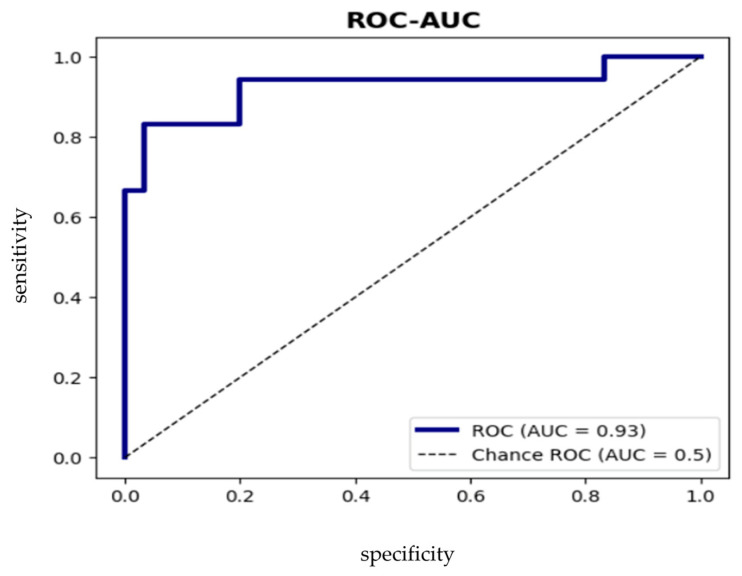
ROC analysis of the model defining the risk of occurrence of obstructive CAD.

**Table 1 ijms-24-17613-t001:** Basic clinical characteristics.

	All CAD (*n* = 64)	INOCA (*n* = 20)	Obstructive CAD (*n* = 44)	Control (*n* = 30)	*p*-Value
Men (%)	33 (51.6)	5 (25)	28 (63.6)	10 (33.3)	0.004 * *p* _INOCA–obstructive CAD_ = 0.012 *p* _obstructive CAD–Control_ = 0.021
Women (%)	31 (48.4)	15 (75)	16 (36.4)	20 (66.7)	
Age (year)	65 [59; 71]	66.5 [62.8; 71.2]	64.0 [56.5; 71]	28.5 [26; 39.2]	<0.001 * *p* _control–INOCA_ < 0.001 *p* _control–obstructive CAD_ < 0.001
BMI (kg/m^2^)	26.7 [24.9; 28.8]	26.20 [25.67; 30.40]	26.23 [24.68; 28.68]	21.95 [20.75; 25.23]	<0.001 * *p* _control–INOCA_ = 0.003 *p* _control–obstructive CAD_ = 0.003
Smoking (%)	9 (14)	3 (15.0)	6 (13.6)	-	0.879
Hemoglobin (g/L)	142 [133; 152]	142 [134; 151]	144 [133; 152]	136 [129; 152]	0.459
Glucose (mmol/L)	5.5 [5.17; 5.8]	5.60 [5.2; 6.21]	5.40 [5.1; 5.63]	4.9 [4.59; 5.35]	<0.001
Creatinine (µmol/L)	89 [78.2; 99.2]	83.4 [74.8; 96.3]	89.80 [81; 101.8]	82 [77.7; 87]	0.106
Total cholesterol (mmol/L)	4.39 ± 1.36	5.11 ± 1.51	3.85 ± 0.95	4.87 ± 0.77	<0.001 * *p* _INOCA– obstructive CAD_ = 0.005 *p* _obstructive CAD_–_Control_ < 0.001
LDL (mmol/L)	2.36 [1.85; 2.97]	2.89 [4.34; 3.62]	2.12 [1.79; 2.48]	2.54 [2.28; 3.21]	<0.001 * *p* _obstructive CAD–INOCA_ < 0.001 *p* _control–obstructive CAD_ = 0.049
HDL (mmol/L)	1.15 [1.02; 1.36]	1.27 [1.06; 1.37]	1.11 [1.02; 1.33]	1.62 [1.35; 1.9]	<0.001 * *p* _control–INOCA_ = 0.015 *p* _control–obstructive CAD_ < 0.001

* Statistically significant *p* < 0.05; *n*—number of patients in the group; CAD—coronary artery disease; INOCA—ischemia with no obstructive coronary arteries; BMI—body mass index; LDL—low density lipoproteins; HDL—high density lipoproteins.

**Table 2 ijms-24-17613-t002:** CAD patients’ therapy characteristics.

	INOCA	Obstructive CAD	*p*-Value
ACE inhibitors	9 (45.0)	27 (62.8)	0.184
ARB II	6 (30.0)	10 (23.3)	0.567
Beta-blocker	17 (85.0)	34 (79.1)	0.737
Calcium channel blockers	11 (55.0)	18 (41.9)	0.33
Antiaggregants	16 (80.0)	40 (93.0)	0.195
Antiarrhythmic drugs	3 (10.3)	6 (20.0)	0.237
HMG-CoA reductase inhibitors	19 (95.0)	44 (100.0)	0.323
Anticoagulants	4 (15)	6 (13.6)	1.000

ACE inhibitors—angiotensin-converting enzyme inhibitors; ARB II—angiotensin II receptor blockers.

**Table 3 ijms-24-17613-t003:** The concentration of WNT proteins in plasma.

Proteins	Groups	Concentration (Me [Q1–Q3])	*p*-Value
LRP6, ng/mL	INOCA	13.02 [12.05–13.7]	0.075
	Obstructive CAD	11.60 [10.5–12.88]	
	Control	12.55 [10.28–14.17]	
WNT1, ng/mL	INOCA	0.15 [0.15–0.16]	<0.001 * *p* _obstructive CAD–INOCA_ < 0.001 *p* _control–INOCA_ = 0.036 *p* _control–CAD_ < 0.001
	Obstructive CAD	0.189 [0.184–0.193]	
	Control	0.15 [0.15–0.184]	
WNT3a, ng/mL	INOCA	0.115 [0.07–0.16]	<0.001 * *p* _obstructive CAD–INOCA_ < 0.001 *p* _control–INOCA_ < 0.001
	Obstructive CAD	0.227 [0.181–0.252]	
	Control	0.25 [0.162–0.37]	
WNT4, ng/mL	INOCA	0.345 [0.278–0.492]	0.015 * *p* _obstructive CAD–INOCA_ = 0.025 *p* _control–_obstructive _CAD_ = 0.047
	Obstructive CAD	0.203 [0.112–0.378]	
	Control	0.345 [0.232–0.528]	
WNT5a, ng/mL	INOCA	0.17 [0.16–0.17]	<0.001 * *p* _obstructive CAD–INOCA_ < 0.001 *p* _control–obstructive CAD_ = 0.001
	Obstructive CAD	0.01 [0.007–0.018]	
	Control	0.16 [0.015–0.17]	
SIRT1, ng/mL	INOCA	1.09 [1.09–1.1]	<0.001 * *p* _obstructive CAD–INOCA_ < 0.001 *p* _control–INOCA_ = 0.012 *p* _control–obstructive CAD_ = 0.007
	Obstructive CAD	0.079 [0.066–0.104]	
	Control	1.09 [0.035–1.1]	

* Statistically significant *p* < 0.05; CAD—coronary artery disease; INOCA—ischemia with no obstructive coronary arteries; LRP6—LDL receptor-related protein 6; SIRT1—sirtuin 1.

**Table 4 ijms-24-17613-t004:** Univariate logistic regression analysis between obstructive CAD and INOCA groups.

Factor/Predictor	B	OR (95%CI) Exp (B) [95%CI]	*p*-Value
miR-21a (REU)	−2.568	0.077 [0.001. 5.757]	*p* = 0.244
miR-145 (REU)	−2.02	0.133 [0.0. 62.492]	*p* = 0.520
miR-221 (REU)	−1.122	0.326 [0.0. 283.87]	*p* = 0.745
LRP6 (ng/mL)	0.475	1.608 [1.037. 2.494]	*p* = 0.034 *
WNT1 (ng/mL)	−1808.16	0.0 [0.0. inf]	*p* = 0.999
WNT3A (ng/mL)	−30.917	0.0 [0.0. 0.00005]	*p* = 0.001 *
WNT4 (ng/mL)	−0.321	0.725 [0.368. 1.429]	*p* = 0.354
WNT5a (ng/mL)	2.466	11.775 [0.298. 465.105]	*p* = 0.188
SIRT1 (ng/mL)	64.169	7.38e + 27 [0.0. inf]	*p* = 0.994
Age (years)	0.036	1.037 [0.969. 1.11]	*p* = 0.297
Smoking (*n*)	0.111	1.117 [0.25. 5.005]	*p* = 0.884
Gender (male/female)	−1.658	0.191 [0.058. 0.622]	*p* = 0.006 *
BMI (kg/m^2^)	0.041	1.042 [0.891. 1.218]	*p* = 0.605
Hypertension (*n*)	0.329	1.39 [0.136. 14.255]	*p* = 0.781
Dyslipidemia (*n*)	0.329	1.39 [0.136. 14.255]	*p* = 0.781
Angina pain (*n*)	0.542	1.719 [0.417. 7.084]	*p* = 0.454
Myocardial infarction (*n*)	−1.923	0.146 [0.03. 0.708]	*p* = 0.017 *
ACE inhibitors	−0.724	0.485 [0.165. 1.422]	*p* = 0.187
ARB II	0.347	1.415 [0.43. 4.647]	*p* = 0.568
Beta blockers	0.405	1.499 [0.359. 6.271]	*p* = 0.579
Calcium channel blockers	0.529	1.697 [0.583. 4.945]	*p* = 0.332
Antiaggregant	−1.204	0.3 [0.06. 1.494]	*p* = 0.142
Statins	−21.683	0.0 [0.0. inf]	*p* = 0.999
Fasting glucose (mmol/l)	−0.021	0.979 [0.898. 1.067]	*p* = 0.635

* Statistically significant *p* < 0.05; CAD—coronary artery disease, INOCA—ischemia with no obstructive coronary arteries; LRP6—LDL receptor-related protein 6; SIRT1—sirtuin 1; BMI—body mass index; ACE inhibitors—angiotensin-converting enzyme inhibitors; ARB II—angiotensin II receptor blockers.

**Table 5 ijms-24-17613-t005:** Multivariate logistic regression analysis between CAD and INOCA groups.

Variables	Coef (B)	Exp (B)	*p*-Value
LRP6, ng/mL	0.451	1.57 [1.17. 2.09]	*p* = 0.002 *
WNT3a, ng/mL	−34.4454	0.0 [0.0. 0.00001]	*p* < 0.001

* Statistically significant *p* < 0.05.

**Table 6 ijms-24-17613-t006:** Primer sequences for RT-PCR.

Primer	Sequence
miR-21a	5′-TAGCTTATCAGACTGATGTTGAAAA-3′
miR-145	5′-TCCAGTTTTCCCAGGAATCCCT-3′
miR-221	5′-GACCTGGCATACAATGTAGATTTAAA-3′

## Data Availability

The data presented in this study are available on request from the corresponding author. The data are not publicly available because some data sets will be used for further research.

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
