# Peer review of "MiRNA-21a, miRNA-145, and miRNA-221 Expression and Their Correlations with WNT Proteins in Patients with Obstructive and Non-Obstructive Coronary Artery Disease"

_ijms, 2023, doi:10.3390/ijms242417613_

Round 1

Reviewer 1 Report

Comments and Suggestions for Authors

See attached PDF for full comments.

Comments on the Quality of English Language

Language is fine. Minor editing required.

Author Response

Dear reviewer!

We sincerely thank you for Your detailed review on our article. Your comments have been taken into account, and we tried to make all the necessary corrections. 

Best regards, Iusupova A.O.

Reviewer 2 Report

Comments and Suggestions for Authors

The manuscript “MicroRNA and WNT signaling cascade proteins in obstructive and non-obstructive coronary artery disease.” represents an interesting original scientific paper describing the role of the miRNAs (miR-21a, miR-145, miR-221) and components of the WNT signaling cascade (WNT1, WNT3a, WNT4, WNT5a) in obstructive CAD and ischemia with no obstructive coronary arteries (INOCA).

However, some major obstacles prevent the paper's acceptance in its present form.

These, among others, include:

Introductory section

The authors should reformulate the introductory section and better describe why they chose to investigate the specified miRNAs and components of the Wnt signaling pathway.

4. Materials and Methods 421

4.1 Patient population

The authors should provide the period in which the patients and control subjects were enrolled in the study, whether any patients were subsequently excluded, and why.

Lines 425-426… The authors have written: “Exclusion criteria are described in the Supplementary.” The authors should provide supplementary materials with specified information.

4.2 Collection of blood samples and ELISA

It would be helpful if the authors could provide the catalog numbers of ELISA kits used in the study. Also, the centrifugation speed (rcf or rpm) parameters should be specified.

There is no information on how the concentration of LRP6 and SIRT1 are measured.

4.3 RNA Extraction and Reverse Transcription-Polymerase Chain Reaction (RT-PCR) Assay

The authors should describe the parameters of the qRT-PCR reaction. Also, the catalog numbers of all primers used in the study should be provided. Did the authors use commercial miScript Primer Assays in all cases? In that case, the catalog numbers should be provided. If not, the way of primer construction or reference source should be provided.

2. Results

Exclusion criteria should be specified in the legend of Figure 1. Also, the text in the bottom box of Figure 1 should be changed (e.g., into Assessment of the level of protein biomarkers and microRNA in blood plasma samples using ELISA and qPCR).

Either Table 1 is missing from the manuscript text or the numerical values of all the tables presented in the manuscript should be changed.

Lines 118-119… The authors have written: “The investigated groups were comparable according to the main clinical and demographic indicators (age, body mass index (BMI)).” However, from the data in Table 2, the groups differ significantly regarding the specified parameters.

Line 132… The authors have written: “The medication therapy did not differ between the groups of patients.” This sentence e should be remodified in line with the previous statement, "… Patients with obstructive CAD, who were treated by higher doses of statins.”

It is unclear what exactly is presented in Tables 5 and 6.- univariate and multivariate analysis of obstructive CAD or INOCA group.

The authors should better describe the selection criteria for multivariate analysis as well.

A major revision of the manuscript is advised.

Author Response

(The authors gave the same response as above.)

Reviewer 3 Report

Comments and Suggestions for Authors

Reviewer report

MicroRNA and WNT signaling cascade proteins in obstructive and non-obstructive coronary artery disease.

This was a very well performed and comprehensive paper that evaluates the levels of various miRNAs in obstructive and non-obstructive coronary artery disease in human blood plasma.

1. What is the global burden of the mentioned diseases.

2.  Can authors mention what is known in the aspect of the current study and what is the novelty of this work.

3.  In brief please explain about biogenesis of miRNA and their role in cardiac regulation.

4.  Can authors explain schematically how Wnt signaling pathway is regulated by miRNAs leading to obstructive and non-obstructive coronary artery disease.

5. Is it possible to report the data from coronary angiography which is mentioned during the study design

6. Please arrange the figure 2 properly as it is overlapping the figure legend.

7. What will be the effect based on the gender variation between male and female.

8. Was there any scope to perform Profile of miRNA and WNT proteins which will be more elaborative in case of samples.

Author Response

(The authors gave the same response as above.)

Round 2

Reviewer 1 Report

Comments and Suggestions for Authors

The authors are to be commended for making all requested edits.

Comments on the Quality of English Language

The language of the revised sections needs quite a lot of work. 

Reviewer 2 Report

Comments and Suggestions for Authors

The authors have successfully revised the manuscript. I have no additional comments or requests.